# Tetramerisation of the CRISPR ring nuclease Crn3/Csx3 facilitates cyclic oligoadenylate cleavage

Januka S Athukoralage, Stuart McQuarrie, Sabine Grüschow, Shirley Graham, Tracey M Gloster*, Malcolm F White*

Biomedical Sciences Research Complex, School of Biology, University of St Andrews, St Andrews, United Kingdom

**Abstract** Type III CRISPR systems detect foreign RNA and activate the cyclase domain of the Cas10 subunit, generating cyclic oligoadenylate (cOA) molecules that act as a second messenger to signal infection, activating nucleases that degrade the nucleic acid of both invader and host. This can lead to dormancy or cell death; to avoid this, cells need a way to remove cOA from the cell once a viral infection has been defeated. Enzymes specialised for this task are known as ring nucleases, but are limited in their distribution. Here, we demonstrate that the widespread CRISPR associated protein Csx3, previously described as an RNA deadenylase, is a ring nuclease that rapidly degrades cyclic tetra-adenylate ($cA_4$). The enzyme has an unusual cooperative reaction mechanism involving an active site that spans the interface between two dimers, sandwiching the $cA_4$ substrate. We propose the name Crn3 (CRISPR associated ring nuclease 3) for the Csx3 family.

*For correspondence:
tmg@st-andrews.ac.uk (TMG);
mfw2@st-and.ac.uk (MFW)

**Competing interests:** The authors declare that no competing interests exist.

## Introduction

The CRISPR system provides adaptive immunity against viruses and other Mobile Genetic Elements (MGE) in bacteria and archaea (reviewed in *Wright et al., 2016*; *Koonin and Makarova, 2017*). Type III CRISPR systems are widespread in archaea and found in many bacteria (*Makarova et al., 2011*), including the human pathogen *Mycobacterium tuberculosis* (*Grüschow et al., 2019*). Type III effectors utilise a Cas7 backbone subunit to bind CRISPR RNA (crRNA) (*Rouillon et al., 2013*). This allows detection of the RNA encoded by invading MGE via base-pairing to the crRNA. Binding of this 'target RNA' results in subtle conformational changes in the effector complex, activating the catalytic Cas10 subunit which uses its HD-nuclease domain to degrade DNA (*Elmore et al., 2016*; *Estrella et al., 2016*; *Kazlauskiene et al., 2016*; *Jung et al., 2015*; *Han et al., 2017b*) and its cyclase domain to synthesise cyclic oligoadenylate (cOA) molecules by polymerisation of ATP (*Kazlauskiene et al., 2017*; *Niewoehner et al., 2017*; *Rouillon et al., 2018*). These cyclic molecules, which range from 3 to 6 AMP monomers in size ($cA_3$ to $cA_6$), act as second messengers in the cell, signalling viral infection and activating cellular defences. $cA_4$ and $cA_6$ bind specifically to two subtypes of CRISPR Associated Rossman Fold (CARF) domain (*Makarova et al., 2014*). These domains are fused to a range of effector domains that carry out defensive duties. The best characterised is the Csx1/Csm6 family that utilises a C-terminal HEPN (Higher Eukaryotes and Prokaryotes, Nucleotide binding) domain to cleave RNA non-specifically (*Niewoehner and Jinek, 2016*; *Sheppard et al., 2016*). The Csx1/Csm6 ancillary ribonucleases are crucial for CRISPR-based immunity in vivo in several different organisms, emphasizing the importance of cOA signalling for type III CRISPR defence (*Jiang et al., 2016*; *Grüschow et al., 2019*; *Foster et al., 2019*; *Deng et al., 2013*). A wide variety of alternative CARF-domain proteins associated with type III CRISPR loci have been identified but not yet described (*Shmakov et al., 2018*; *Shah et al., 2019*), and recent work

**eLife digest** Bacteria protect themselves from infections using a system called CRISPR-Cas, which helps the cells to detect and destroy invading threats. The type III CRISPR-Cas system, in particular, is one of the most widespread and efficient at killing viruses.

When a bacterium is infected, the CRISPR-Cas system takes a fragment of the genetic material of the virus, and copies it into a molecule. These molecular 'police mugshots' are then loaded into a complex of Cas proteins that patrol the cell, looking for a match and destroying any virus that can be identified.

Some Cas proteins also produce alarm signals, called cyclic oligoadenylates (cOAs), which can trigger additional defences. However, this process can damage the genetic material of the bacterium, harming or even killing the cell.

Enzymes known as ring nucleases can promptly degrade cOAs and turn off this defence system before it causes harm. However, ring nucleases have only been found in a few species to date; how most bacteria deal with cOA toxicity has remained unknown. Here, Athukoralage et al. set out to determine whether a widespread enzyme known as Csx3, which is often associated with type III CRISPR-Cas systems, could be an alternative off switch for cOA triggered defences.

Initial 'test tube' experiments with purified Csx3 proteins confirmed that the enzyme could indeed break down cOAs. A careful dissection of Csx3's molecular structure, using biochemical and biophysical techniques, revealed that it worked by 'sandwiching' a cOA molecule between two co-operating portions of the enzyme. As a final test, Csx3 was introduced into strains of bacteria genetically engineered to have a fully functional Type III CRISPR-Cas system. In these cells, Csx3 successfully turned off the Type III immune response.

These results reveal a new way that bacteria avoid the toxic side effects of their own immune defences. Ultimately, this could pave the way for the development of anti-bacterial drugs that work by blocking Csx3 or similar proteins.

has characterized the $cA_4$-activated DNA nickase Can1, which is present in *Thermus thermophilus* (**McMahon et al., 2020**).

Thus, type III CRISPR systems are capable of directing a multi-faceted antiviral defence on detection of foreign RNA in the cell. However, activation of this anti-viral state is known to generate collateral damage to host nucleic acids (**Rostøl and Marraffini, 2019**) that could lead to dormancy or cell death. While this could be an acceptable outcome for an infected cell, if a viral infection can be cleared then the cOA-signalling pathway needs a mechanism to remove the cOA molecules and return the cell to a basal state. Some archaea encode a dedicated ring nuclease (**Athukoralage et al., 2018**), recently named as the Crn1 family (CRISPR-associated ring nuclease 1) (**Athukoralage et al., 2020a**). Crn1 is a specialised CARF domain protein that splits the $cA_4$ ring into two linear $A_2$ products to switch off the antiviral response (**Athukoralage et al., 2018**). In other systems, the CARF domains of Csx1/Csm6 family nucleases slowly degrade cOA, thus acting as bi-functional, self-limiting nucleases (**Athukoralage et al., 2019**; **Jia et al., 2019**; **Garcia-Doval et al., 2020**). Many archaeal viruses and some bacteriophage also encode a specialised ring nuclease as an anti-CRISPR (Acr). This enzyme, known as AcrIII-1, binds $cA_4$ by means of a distinct protein fold (DUF1874) unrelated to the CARF domain, and degrades $cA_4$ rapidly using conserved active site residues (**Athukoralage et al., 2020a**). These viral ring nucleases can abrogate type III CRISPR immunity by rapidly destroying the $cA_4$ infection signal. This enzyme has been co-opted into some bacterial type III CRISPR systems, where it likely acts to degrade $cA_4$ following clearance of phage infection. In this context, it has been named as CRISPR-associated ring nuclease 2 (Crn2) (**Athukoralage et al., 2020a**; **Samolygo et al., 2020**).

Ring nucleases thus appear to be an important constituent of virus:host conflict in the expanding arena of cyclic nucleotide signalling. Here, we focus on the Csx3 family of proteins, which is found associated with many type III CRISPR systems (**Shah et al., 2019**; **Shmakov et al., 2018**). Csx3 from *Archaeoglobus fulgidus* has been described as an RNA deadenylation enzyme and crystallised in complex with a pseudo-symmetric RNA tetraloop (**Yan et al., 2015**). Subsequent structural analysis suggested that the Csx3 protein may be a distantly related member of the CARF family of proteins

(*Topuzlu and Lawrence, 2016*). Spurred by these observations, we undertook a detailed study of the Csx3 protein. We demonstrate that Csx3 is, in fact, a $cA_4$-specific ring nuclease, for which we propose the name Crn3 (CRISPR-associated ring nuclease 3). The enzyme has an unusual cooperative reaction mechanism involving the association of two dimers, bridged by the $cA_4$ substrate, to complete the active site. This mechanism may allow the cell to respond appropriately to changing $cA_4$ and Csx3 levels during viral infection and preserve type III CRISPR immunity.

## Results

### Csx3 is a ring nuclease specific for binding and cleavage of $cA_4$

The Csx3 protein from *A. fulgidus* was originally crystallised in the absence and presence of a pseudo-symmetric RNA tetranucleotide, and shown to have RNA deadenylase activity in vitro (*Yan et al., 2015*). Given what is now known about $cA_4$ ring nucleases, we re-assessed the structure and potential function of the Csx3 protein. It has previously been suggested that the binding site for the RNA tetranucleotide could be compatible with binding of a symmetric cyclic oligonucleotide (*Topuzlu and Lawrence, 2016*), similar to those observed in other CARF family proteins. Examination of the genomic context of Csx3 in selected archaeal and bacterial species confirmed its close association with type III-B CRISPR systems and with CARF family proteins such as the ribonuclease Csx1 (*Figure 1A*), implying a role in cyclic oligoadenylate signalling and providing further impetus to re-examine the function of Csx3. Csx3 is most commonly observed in the euryarchaea and cyanobacteria (*Figure 1—figure supplement 1*). We therefore cloned, expressed and purified *A. fulgidus* Csx3, allowing biochemical analysis. Initial studies showed that Csx3 binds cyclic tetra-adenylate with an apparent dissociation constant <0.1 μM. In contrast, an RNA oligonucleotide (49-9A) with a 9A poly-adenylate 3' tail bound 100-fold less tightly, with an apparent dissociation constant around 10 μM (*Figure 1B*).

The enzymatic specificity of Csx3 was tested against an RNA substrate oligonucleotide 49-9A. Csx3 cleaved this substrate in the presence of $Mn^{2+}$ ions, removing nucleotides from the 3' end

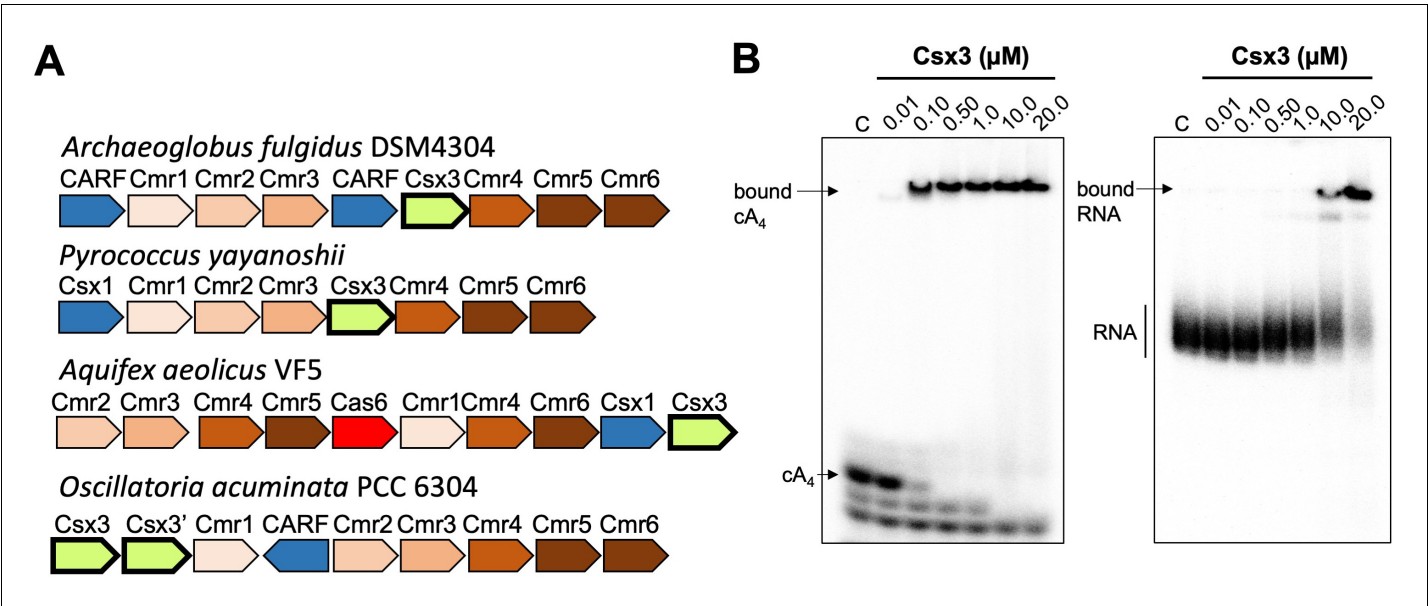

**Figure 1.** Csx3 is a type III CRISPR accessory protein that binds $cA_4$. (**A**) Genome context of selected Csx3 orthologues. Csx3 is found next to type III-B CRISPR operons and adjacent to Csx1 and other uncharacterised CARF family proteins. Csx3' is a longer version of Csx3, common in cyanobacteria, consisting of an N-terminal Csx3 domain fused to a C-terminal kinase/transferase domain of unknown function. (**B**) Phosphor images of native gel electrophoresis visualising $cA_4$ (20 nM) or RNA oligonucleotide 49-9A (50 nM) binding by *A. fulgidus* Csx3. Csx3 binds to $cA_4$ with high affinity (apparent $K_D$ <100 nM) and binds the RNA 49-9A with significantly lower affinity (apparent $K_D$ ~10 μM). Images are representative of three technical replicates.

The online version of this article includes the following figure supplement(s) for figure 1:

**Figure supplement 1.** Multiple sequence alignment of Csx3 proteins showing conserved residues.

(*Figure 2A*), in keeping with previous observations (*Yan et al., 2015*). However, Csx3 cleaved $cA_4$ with a far higher rate (*Figure 2B*). The single-turnover rate constants were determined as $0.0063 \pm 0.0013$ min$^{-1}$ for RNA and $3.5 \pm 0.16$ min$^{-1}$ for $cA_4$ (*Figure 2C*). The ~600 fold faster cleavage rate for $cA_4$ compared to RNA strongly suggests that $cA_4$ is the physiological substrate, and

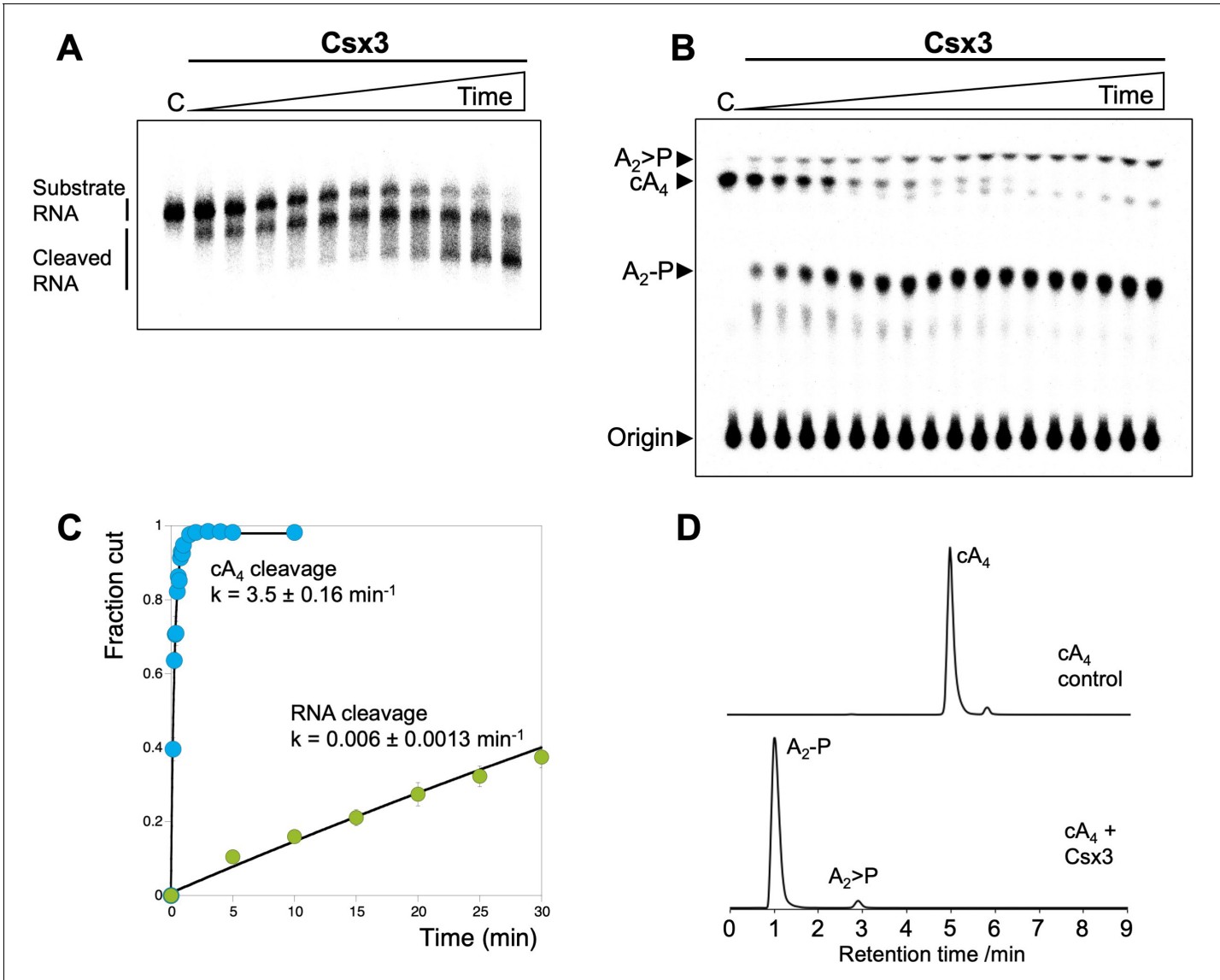

**Figure 2.** Csx3 is a potent ring nuclease. (A) Denaturing polyacrylamide gel electrophoresis visualising cleavage of 5'-end radiolabelled RNA 49-9A (50 nM) by Csx3 (8 µM dimer) at 50°C (time points every 5 min from 5 to 40 min, then 50, 60 and 90 min, n = 3 technical replicates). (B) TLC analysis of the reaction products of radiolabelled $cA_4$ (200 nM) incubated with *A. fulgidus* Csx3 (8 µM dimer) in reaction buffer at 50°C (time points every 5 s from 10 to 60 s then 1.5, 2, 3, 4, 5 and 10 min, n = 6 technical replicates). The $cA_4$ was rapidly converted into $A_2$-P, with a small amount of $A_2$ >P (linear $A_2$ with a cyclic 2',3' terminal phosphate). (C) Plot comparing single-turnover kinetics of $cA_4$ (blue) and RNA 49-9A (green) cleavage by AfCsx3 at 50°C. Data points are the average of three technical replicates and error bars represent the standard deviation of the mean. (D) Liquid-chromatography high-resolution mass spectrometry analysis of reaction products when $cA_4$ (100 µM, top) was incubated with AfCsx3 (10 µM) for 10 min at 50°C (bottom). $cA_4$ was fully degraded to form $A_2$-P (di-adenylate containing a 3' phosphate) with a minor amount of $A_2$ >P (di-adenylate containing a 2',3' cyclic phosphate).

The online version of this article includes the following source data and figure supplement(s) for figure 2:

**Source data 1.** Excel spreadsheet with raw data.
**Source data 2.** Excel spreadsheet with raw data.
**Figure supplement 1.** Csx3 from *M. mazei* is a ring nuclease.

that RNA with a 3′ polyA tail represents a substrate analogue of the cyclic nucleotide, as observed previously for a Csx1/Csm6 family enzyme (*Han et al., 2017a*). Although the physiological growth temperature of *A. fulgidus* is around 70°C, we conducted these assays at 50°C to enable rate determination.

The products of the $cA_4$ cleavage reaction were identified by liquid-chromatography high-resolution mass spectrometry (LC-HRMS) as predominantly linear di-adenylate ($A_2P$), plus a small amount of $A_2P$ with a cyclic phosphate ($A_2 > P$) (*Figure 2D*). This suggests there are two active sites in the dimeric Csx3 structure, as seen for the ring nuclease Crn1 (*Athukoralage et al., 2018*). Similar results were obtained for Csx3 from the mesophilic archaeon *Methanosarcina mazei* (*Figure 2—figure supplement 1*).

## Csx3 can decrease $cA_4$-mediated immunity in vivo

To determine whether Csx3 displayed properties consistent with a function as a ring nuclease in vivo, we made use of the synthetic *M. tuberculosis* (Mtb) type III CRISPR system that we recently established in *E. coli* (*Grüschow et al., 2019*). This system allows expression of the Mtb Csm complex, defined CRISPR RNA, and the processing enzyme Cas6 along with a cOA effector protein of choice. In this case, we used the previously characterised Csx1 ribonuclease from *Thioalkalivibrio sulfidiphilus*, which is activated by $cA_4$ (*Grüschow et al., 2019*). Csx1 provided significant immunity (three logs) against plasmid transformation when a targeting crRNA specific for the plasmid was provided (*Figure 3*). As a control, we added the phage ring nuclease AcrIII-1 from bacteriophage THSA-485A, which we previously showed could abolish immunity mediated by $cA_4$ (*Athukoralage et al., 2020a*), and observed the expected loss of plasmid targeting, reflected in higher transformation efficiencies. When the phage ring nuclease gene was replaced by a gene encoding Csx3 from *M. mazei* (chosen as the organism grows at close to 37°C, in common with *E. coli*) we observed smaller but significant increases in plasmid transformation efficiency, with an effect that increased from 18 to 40 hr growth post-transformation. These observations were consistent with a partial deactivation of the Csx1 ribonuclease due to degradation of $cA_4$ by the Csx3 enzyme. The effect is not as striking as when using a *bona fide* Acr protein, which is consistent with the prediction that Csx3 functions as part of the type III CRISPR defence in cells that express it – rather than an Acr. It should be noted that we expressed the Csx3 enzymes using a strong inducible promoter in these assays and we do not know what the relevant Csx3 concentrations are in virally-infected cells.

## Essential catalytic residues are widely separated in Csx3

Previously, the RNA cleavage activity of *A. fulgidus* Csx3 was shown to be dependent on the presence of manganese ions. The H60 residue was shown to be essential for RNA cleavage, as an H60A variant was inactive, while the H57A and H80A variants showed reduced activity (*Yan et al., 2015*). As these residues are positioned on the opposite surface of the dimer from the RNA binding site (*Figure 4A*), this led to the prediction that the RNA binding site and RNA cleavage site existed on opposing sides of the dimeric structure, which was plausible for an RNA substrate (*Yan et al., 2015*). We tested the metal ion dependence of the $cA_4$ cleavage activity and confirmed that the presence of $Mn^{2+}$ or $Co^{2+}$ ions was required for catalysis (*Figure 2—figure supplement 1* and *Figure 4—figure supplement 1*). We recapitulated the H60A variant and found that, in agreement with the previous study, the variant lacked any detectable catalytic activity (*Figure 4B*). Thus, it appears that H60 and the presumed metal binding site are essential for the ring nuclease activity of Csx3 despite being situated on the opposite face of the Csx3 dimer, over 20 Å away from the $cA_4$ binding site.

Examination of the multiple sequence alignment of Csx3 (*Figure 1—figure supplement 1*) revealed the presence of a conserved aspartate residue, D69, which is positioned adjacent (~4 Å) to the bound RNA in the crystal structure (*Figure 4A*). The importance of D69 was not explored in previous studies, so we mutated D69 to an alanine to test for a role in catalysis. The D69A variant was completely catalytically inactive as a ring nuclease, confirming the importance of D69 in the catalytic mechanism and suggesting that catalysis requires residues on both faces of the dimer (henceforth denoted as the 'D69 face' and 'H60 face'). A possible explanation for this was that Csx3 has a shared active site that is formed when two dimers come together, forming a tetramer. A diagnostic test for this, as first proposed for the shared active site of aspartate transcarbamoylase (*Wente and*

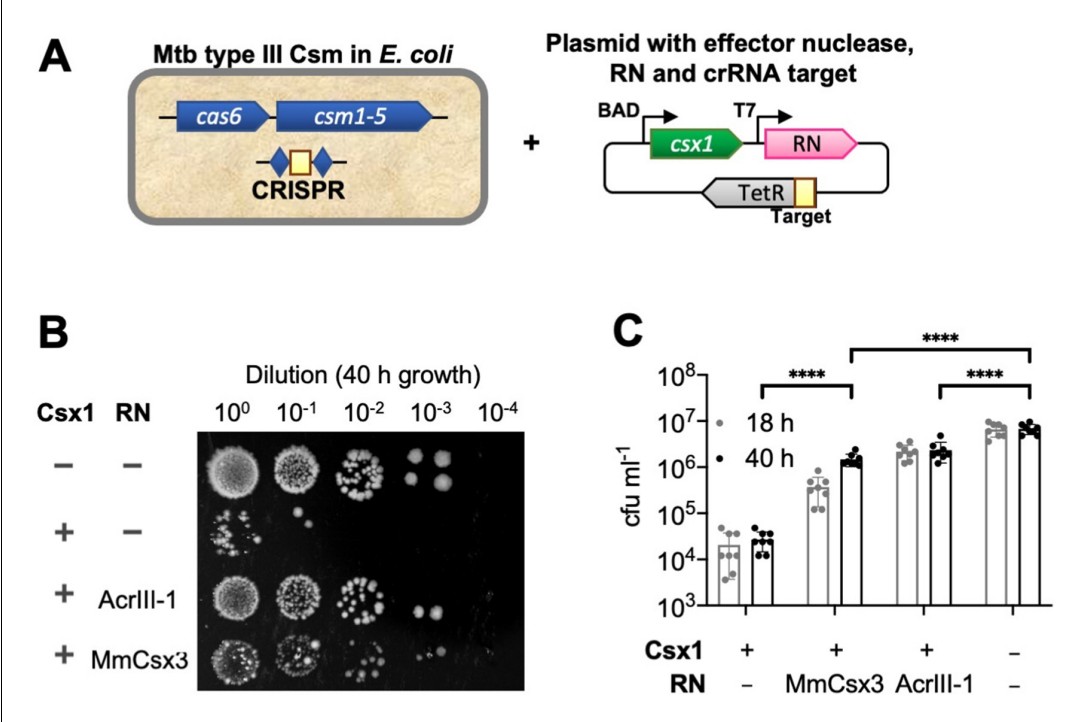

**Figure 3.** Csx3 functions as a ring nuclease in vivo. (**A**) Schematic of plasmid transformation assay. A plasmid containing the effector nuclease Csx1, a ring nuclease (RN) and a target sequence that is complementary to the crRNA is transformed into *E. coli* expressing the *M. tuberculosis* (Mtb) Csm complex charged with crRNA. (**B**) The effector ribonuclease Csx1 prevents an incoming plasmid carrying a CRISPR-target sequence from being established, resulting in a reduction of number of transformants by almost 3 orders of magnitude under selective conditions. Both AcrIII-1 from phage THSA-485 and *M. mazei* (Mm)Csx3 provide a level of protection against Csx1-mediated plasmid immunity, with MmCsx3 requiring an extended incubation period to achieve a similar effect as its viral counterpart. (**C**) Number of transformants of Csx1-mediated plasmid immunity in the absence and presence of $cA_4$-degrading ring nucleases. AcrIII-1 almost completely prevented Csx1-mediated cell death or growth arrest, whereas MmCsx3 yielded partial but still significant relief from immunity. Transformants were counted after 18 and 40 hr of growth. Data are representative of two biological replicates and four technical replicates each, N = 8; individual data points are shown and error bars represent the standard deviation. Significance threshold was set at $p<0.05$, \*\*\*\*: $p<0.00001$ (unpaired, two-tailed Welch t test). Key: RN – ring nuclease; MmCsx3 – *M. mazei* Csx3.

*Schachman, 1987*), is to mix inactive single variants and look for recovery of activity, as a fraction of intact active sites can be formed in the quaternary structure. Accordingly, we took the two inactive variants of Csx3, H60A and D69A, and tested their ring nuclease activity when mixed together (*Figure 4B*). Ring nuclease activity was recovered when the two inactive variants were combined. This result was strongly supportive of the hypothesis that the reaction mechanism involves two half-sites that combine to form a single active site that bridges two dimers of the enzyme.

## The structure of Csx3 reveals a head-to-tail filament stabilised by $cA_4$

In parallel with the biochemical analysis, we crystallised the H60A variant of Csx3 and soaked the $cA_4$ substrate into the crystals. The structure was solved using molecular replacement with data to 1.84 Å resolution (*Supplementary file 1*). The electron density clearly showed a molecule of $cA_4$ bound between adjacent dimers of Csx3 (*Figure 5A*). Notably, the Csx3 structure presented here crystallised in a different space group to those structures published previously (*Yan et al., 2015*), which has allowed the arrangement of Csx3 dimers into a pseudo filament arrangement (*Figure 5—figure supplement 1*) with a $cA_4$ molecule sandwiched between the D69 and H60 faces of the protein. The rotation between adjacent dimers in the filament is 144 °, coupled with a translation of 28 Å. The calculated buried surface between the complex arrangement of protein and ligand reflects the number of interactions formed. The interface of the two monomers forming the dimer is around 1020 Å². The interface area between the D69 face, which forms the majority of the interactions with $cA_4$, is around 940 Å² (470 Å² per monomer), and for the H60 face is around 340 Å² (170 Å² per monomer). Interestingly, although not annotated as a tetramer, there is a buried surface area of

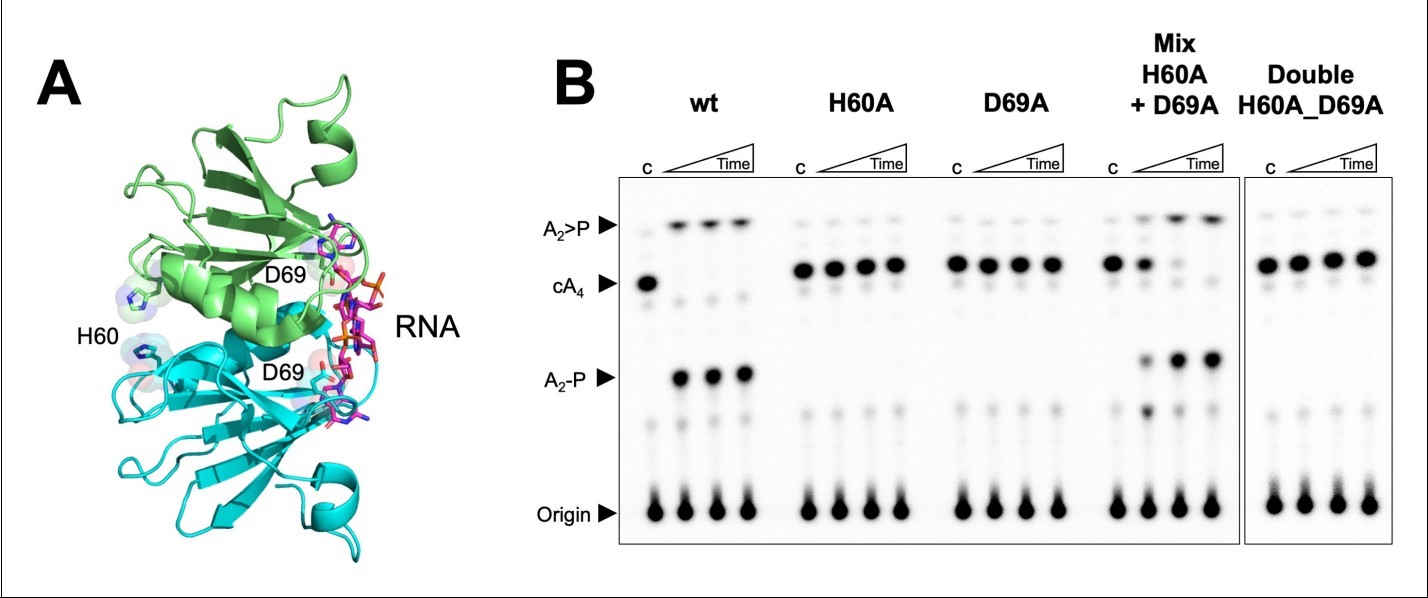

**Figure 4.** Both faces of the Csx3 dimer are required for ring nuclease activity. (**A**) Structure of the Csx3 dimer (monomers in cyan and green) bound to an RNA tetraloop (magenta) (PDB 3WZI), with residues D69 and H60 indicated. (**B**) Phosphorimage of TLC visualising $cA_4$ (~200 nM) cleavage by A. fulgidus Csx3 (8 μM protein dimer) wild-type and variants at 70°C. While neither the H60A nor the D69A variant of Csx3 had detectable ring nuclease activity, a mixture of the two inactive variants restored activity. The double mutant was inactive. Control lane c – $cA_4$ in absence of protein. Time points were 10, 60 and 600 s. This phosphorimage is representative of 3 technical replicates.

The online version of this article includes the following figure supplement(s) for figure 4:

**Figure supplement 1.** Metal dependence of Csx3.

around 700 $\text{Å}^2$ (350 $\text{Å}^2$ per monomer) between the two adjacent dimers. It is difficult to rationalise that the structure of Csx3 in complex with $cA_4$ formed following overnight soaks of the compound with apo crystals, given the size of $cA_4$ and rigidity of crystal packing. From comparison to the apo Csx3 structure, we hypothesize there are no significant conformational changes or large loop movements upon binding, but instead suggest subtle movements of active site and interacting residues to accommodate $cA_4$.

There are surprisingly few residues that interact with $cA_4$ in the active site of Csx3, given the size of the ligand. The orientation of each monomer in the dimer means the interactions with $cA_4$ are symmetrical. The majority of interactions are made by the D69 face. The $cA_4$ hydrogen bonds with the main chain atoms of I22, S44, G45, and I49, and side chain atoms of S44, R46 and R71 (*Figure 5—figure supplement 2*). Despite the number of main chain interactions, these residues are either absolutely (G45, I49, and R71) or highly (I22, S44 and R46) conserved (*Figure 1—figure supplement 1*). The one interaction evident from the H60 face with $cA_4$ is a strong (2.4–2.5 Å) hydrogen bond between H80 and the 2'-OH of the ribose ring. H60, H80, and another histidine residue (H57) nearby on the H60 face are all absolutely conserved in the Csx3 family (*Figure 5B,C*; *Figure 1—figure supplement 1*). The ribose of each AMP moiety of $cA_4$ adopts a relaxed 2'-endo conformation, with the four adenine bases each filling a pocket. Given the Csx3 structure in complex with an RNA fragment (*Yan et al., 2015*) had two adenine, one uracil and one guanine bases in the same position as the four adenine bases described here, there is obviously some plasticity around these recognition sites.

We anticipated that the comparison of the structure of Csx3 with $cA_4$ with the apo structure published by *Yan et al., 2015*, in conjunction with activity assays on Csx3 variants, might provide clues as to the key residues in catalysis. Interestingly, R71 moves around 3.3 Å in order to form bidentate hydrogen bonds with an oxygen atom of a phosphate group, which we predict is adjacent to the phosphodiester bond that is cleaved (*Figure 5C* and text below). This movement of R71 brings it within hydrogen bonding distance of D69, which has been shown to be vital for activity (*Figure 5—figure supplement 3*). The position of H80 also differs between the two structures; the residue

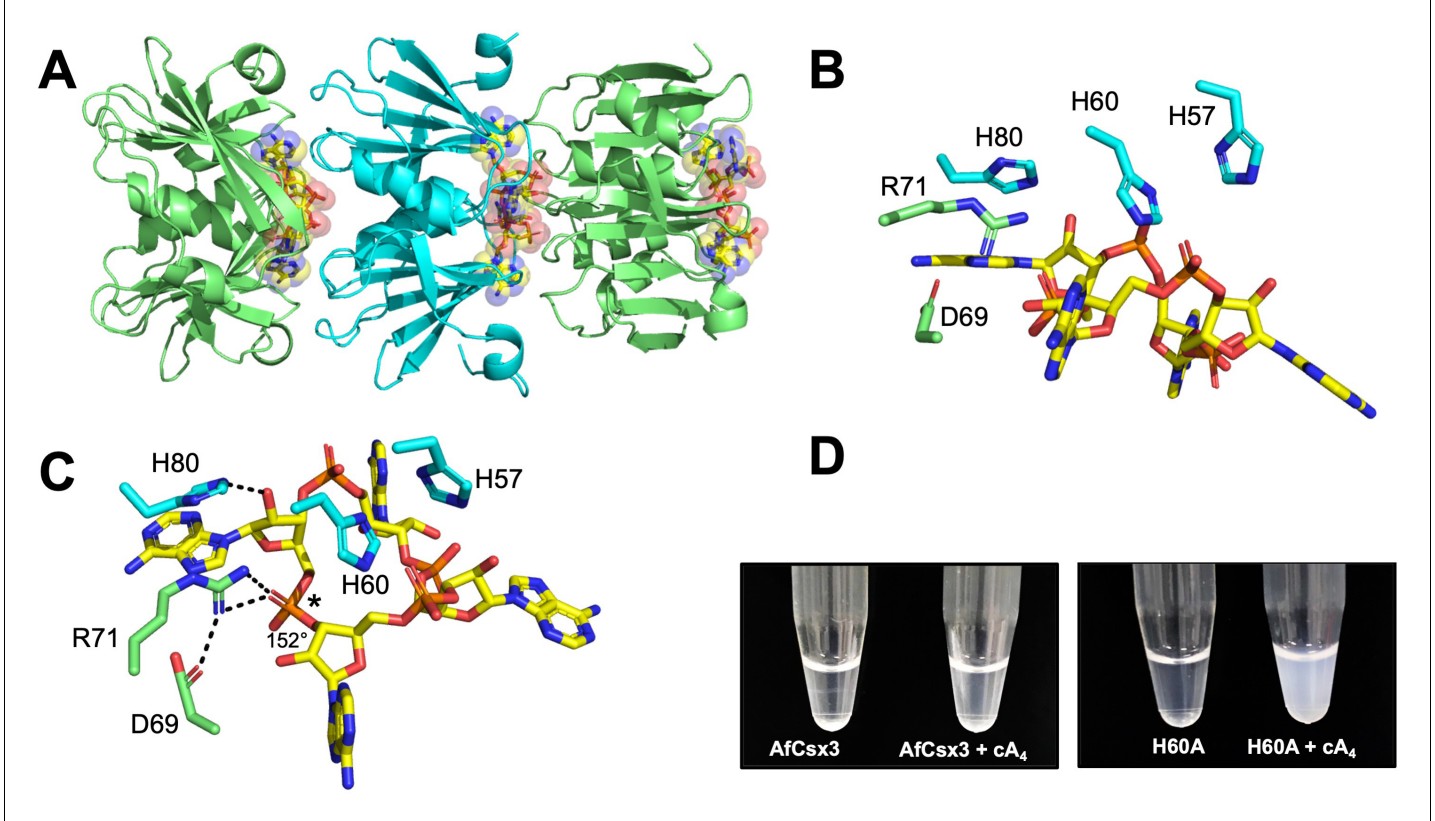

**Figure 5.** Structure of the H60A variant of Csx3 in complex with $cA_4$. (**A**) Crystal structure of H60A Csx3 in complex with $cA_4$. The crystal lattice reveals an extended filament with dimers of Csx3 sandwiching $cA_4$. Three dimers are shown (green and cyan), with the $cA_4$ in yellow sticks. (**B** and **C**) Two views of the Csx3 H60A variant in complex with $cA_4$ showing the conserved residues implicated in binding and catalysis (colouring as in panel A). Whilst H60 was not present in our structure (as an alanine variant was crystallised), the position has been inferred by superposition with PDB 3WZI. In panel C, hydrogen bonds are indicated as black dashed lines. The geometry of the phosphodiester bond is labelled (*), which is likely positioned by a bidentate hydrogen bond with R71, meaning it is suitable for in-line nucleophilic attack by the adjacent 2'-OH moiety. (**D**) Photograph comparing wild-type and H60A Csx3 (80 µM dimer) in the absence and presence of equimolar $cA_4$. The inactive variant forms a milky, colloidal liquid when incubated with $cA_4$, consistent with the formation of extended fibres.

The online version of this article includes the following figure supplement(s) for figure 5:

**Figure supplement 1.** The crystal lattice of the Csx3:$cA_4$ complex.

**Figure supplement 2.** Schematic showing interactions between Csx3 and $cA_4$.

**Figure supplement 3.** Superimposition of Csx3 in complex with $cA_4$ and apo Csx3.

**Figure supplement 4.** Histidine residues in the active site of Csx3.

**Figure supplement 5.** Investigation of H80A and R71A variants of Csx3.

moves around 3.3 Å in order to hydrogen bond with the $cA_4$. However, there is the caveat that the packing arrangement differs in the apo Csx3 structure, where H80 does not interact with a ligand or the face of an adjacent dimer and thus has nothing to 'anchor' it in place.

There are a number of histidine residues in or near the active site which could be involved in coordinating one or more $Mn^{2+}$ ions (*Figure 5—figure supplement 4*). H60 and H57 are the most likely candidates; they are both absolutely conserved, and the position of the alanine residue in the H60A variant structure suggests this residue does not move significantly upon tetramer formation. The position of H57 is identical in the apo and $cA_4$-bound structures. H60 and H57 are at the symmetry plane for the monomers constituting a dimer, meaning there are four histidine residues in close proximity. It is therefore feasible that one or both H60 and/or H57 residues coordinate one or more metal ions, and there is the possibility that just one metal ion bridges the two monomers.

## Predicted mechanism of Csx3

The structure of the Csx3:cA$_4$ complex reveals further details of the catalytic mechanism employed by the enzyme. The other ring nucleases characterised to date are metal independent enzymes that are predicted to work by catalysing in-line nucleophilic attack by a 2'-hydroxyl group of the cA$_4$ substrate on an adjacent phosphodiester bond. In this regard, the best candidate for cleavage is the P-O bond of the phosphate (labelled *), as the angle formed between the 2'-OH, P and O moieties is 152° (*Figure 5C*), which although lower to that seen for the ring nuclease AcrIII-1 (*Athukoralage et al., 2020a*), is still significantly higher than the angle (129°) between the same atoms in both adjacent adenine moieties. The absolutely conserved residue R71 interacts with this phosphate group via a bidentate hydrogen bond and may participate in stabilisation of the transition state or oxyanion leaving group. The conserved catalytic residue D69 in turn makes polar contacts with R71. These two residues may serve to position each other correctly (and/or perturb their respective p$K_a$'s) to enhance catalysis. The conserved H80 residue forms a hydrogen bond with the cA$_4$. As it is one of only two residues from the H60 face that interacts with cA$_4$, H80 may play a 'pinning' role to ensure engagement of the H60 face in catalysis. Csx3 variants R71A and H80A both displayed highly reduced ring nuclease activity (*Figure 5—figure supplement 5*), consistent with important roles in cA$_4$ binding and/or catalysis.

The observations that Mn$^{2+}$ is required for catalysis, and that the primary product of cA$_4$ cleavage is A$_2$P rather than A$_2$>P (*Figure 2—figure supplement 1* and *Figure 4—figure supplement 1*) are suggestive (but not diagnostic) of a mechanism whereby a metal-activated hydroxyl ion initiates nucleophilic attack on the phosphodiester bond (*Yang, 2011*). Uncertainty arises from the observation that the metal independent ring nuclease AcrIII-1 also largely generates A$_2$P, presumably due to the rapid hydrolysis of the cyclic phosphate following phosphodiester bond cleavage (*Athukoralage et al., 2020a*). The conserved histidine residues H57 and H60 are in appropriate positions to contribute to catalysis by coordination of the essential catalytic Mn$^{2+}$ ion(s). We see no evidence for the ion in our crystal structure, which may be due to the deletion of the H60 side chain. However, we note that the five Mn$^{2+}$ ions modelled in the Csx3 structure by Yan et al. have ambiguous electron density, hampered by the lower resolution data (*Yan et al., 2015*). In addition, the coordination distances for a Mn$^{2+}$ ion with a nitrogen atom (of the histidine residues) are longer than expected (*Zheng et al., 2017*). It remains possible that H80 participates in binding a second metal ion in addition to its observed interaction with cA$_4$.

The crystal structure of the Csx3:cA$_4$ complex neatly explains the observation of two distant active site regions, which come together in the complex. During the catalytic cycle, cA$_4$ binding and dimer:dimer sandwiching would lead to rapid cA$_4$ degradation that presumably in turn results in dissociation of the tetrameric active form. In support of this model, we observed that the inactive H60A variant of Csx3 had a milky, colloidal property on addition of cA$_4$, which we interpreted as being due to the formation of long Csx3 fibres bridged by multiple cA$_4$ molecules (*Figure 5D*). The wild-type protein did not show this behaviour, presumably because cA$_4$ was rapidly cleaved. We confirmed the propensity of Csx3 to oligomerise upon cA$_4$ addition by dynamic light scattering (DLS) measurements in buffer devoid of metal ions. Under these conditions, the addition of cA$_4$ resulted in mixed oligomeric species with significantly increased particle size and molecular weight, whereas the molecular weight of particles formed in the absence of cA$_4$ were consistent with homodimers of Csx3 (*Supplementary file 2*).

The discovery that two dimers of Csx3 must associate to sandwich the cA$_4$ substrate to effect cleavage opens the possibility of cooperative kinetic control in the Csx3 reaction cycle. To investigate this, we measured the initial reaction velocity of Csx3 ring nuclease activity under conditions of saturating cA$_4$ at a range of enzyme concentrations. A plot of initial velocity against [E] revealed a sigmoidal relationship that could be fitted to the Hill equation with a Hill coefficient of 1.8 (*Figure 6*), consistent with obligate dimerization (of Csx3 dimers) in the reaction cycle. Practically, this cooperativity may be important to modulate the ring nuclease activity of Csx3 appropriately in the cell, particularly if the concentration of Csx3 changes in response to viral infection. The maximal multiple turnover rate constant observed under these conditions was 0.075 min$^{-1}$ at 70°C, about 50-fold lower than the rate for the chemical step of catalysis measured under single turnover conditions at 50°C. Thus, the association of Csx3 dimers sandwiching cA$_4$, or the dissociation of the complex following product formation, limits the turnover number of the enzyme significantly.

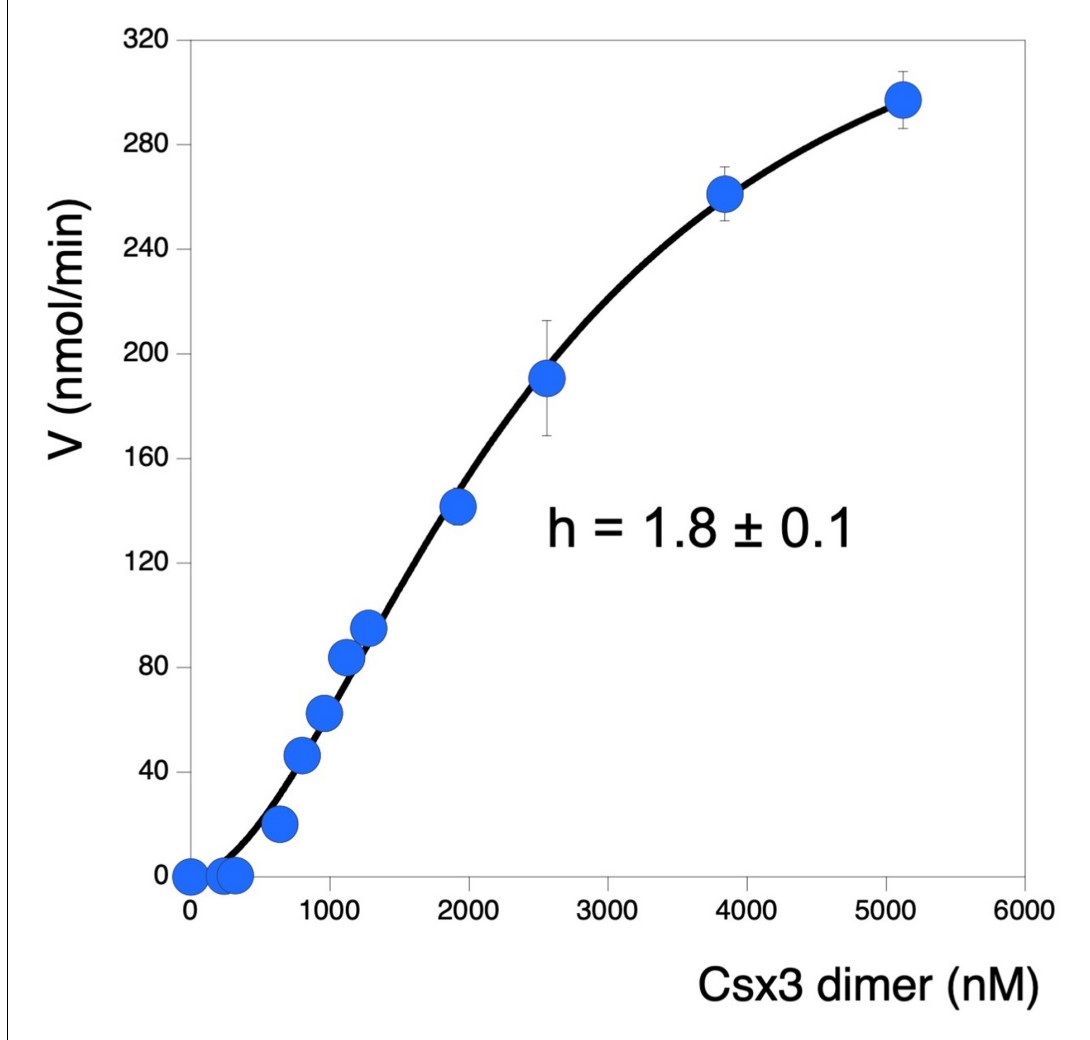

**Figure 6.** Sigmoidal response of ring nuclease activity as a function of Csx3 concentration. Plot visualising initial reaction rates of $cA_4$ cleavage across increasing Csx3 concentrations. The $cA_4$ concentration was 129 μM (25-fold greater than the highest concentration of protein assayed) for all experiments. Instead of a linear relationship between enzyme concentration and activity, a pronounced sigmoidal shape was observed. This could be fitted to a Hill equation to give a good fit with a Hill coefficient of 1.8, consistent with a requirement for two Csx3 dimers to associate with one $cA_4$ substrate molecule to effect catalysis. The maximum multiple turnover rate constant observed was 0.075 $min^{-1}$ at 70°C. Each datapoint is the average of three technical replicates and error bars show the standard deviation of the mean.

The online version of this article includes the following source data for figure 6:

**Source data 1.** Excel spreadsheet with raw data.

## Discussion

### Csx3 is a $cA_4$-specific ring nuclease, Crn3

Here we re-examined the specificity of the Csx3 nuclease, which is found in association with type III CRISPR systems in bacteria and archaea. Csx3 was originally described as a manganese dependent RNA deadenylase (*Yan et al., 2015*). The enzyme was co-crystallised with an RNA tetranucleotide, revealing a pseudo-symmetrical binding mode, and site directed mutagenesis revealed a crucial role for residues including H60 in catalysis – a residue that was situated on the opposite face of the protein from the RNA binding site (*Yan et al., 2015*). This prompted the authors to make the reasonable speculation that RNA is bound on one side and wraps around the protein to be cleaved on the opposite side. Subsequently, Csx3 was identified as a divergent member of the CARF domain protein family (*Topuzlu and Lawrence, 2016*) that bind cyclic oligoadenylates generated by the Cas10

subunit of type III CRISPR systems (*Niewoehner et al., 2017*; *Kazlauskiene et al., 2017*). Coupled with the lack of an obvious role for a RNA deadenylase in type III CRISPR defence, this led us to speculate on its function in vivo. While we confirmed the main experimental observations of the original study (*Yan et al., 2015*), our analyses have revealed that Csx3 binds much more tightly to $cA_4$ than to RNA, is considerably more active as a $cA_4$-specific ring nuclease than an RNA deadenylase, and has properties consistent with a function as a ring nuclease in vivo. These data indicate that Csx3 functions as a $cA_4$-specific ring nuclease in type III CRISPR systems. The default 'Csx' nomenclature was designed for proteins of unknown function lacking a clear association with a specific CRISPR subtype (*Haft et al., 2005*). We therefore propose the family name Crn3 (CRISPR-associated ring nuclease 3) for the Csx3 protein family.

The Crn1 family of ring nucleases, based on the CARF domain fold, was originally identified in the *Sulfolobales* and related crenarchaea (*Athukoralage et al., 2018*), where their function is thought to be to remove $cA_4$ from the cell once a viral infection has been cleared. The anti-CRISPR ring nuclease AcrIII-1 appears in many archaeal virus and some bacteriophage genomes (*Athukoralage et al., 2020a*). Homologues of AcrIII-1 are also found associated with some bacterial type III CRISPR systems, and in this context have been named as CRISPR associated ring nuclease 2 (Crn2) enzymes (*Athukoralage et al., 2020a*; *Samolygo et al., 2020*). Ring nuclease activity has also been identified associated with the CARF domains of some Csx1/Csm6 family ribonucleases, which are therefore bifunctional, self-limiting CRISPR ancillary defence enzymes (*Jia et al., 2019*; *Athukoralage et al., 2019*; *Garcia-Doval et al., 2020*). Overall, the emerging picture is becoming clearer for the specific ring nucleases as they are identified in different archaeal and bacterial genomes (*Figure 7*). It appears to be an emerging paradigm that cells with a type III CRISPR defence require a mechanism to remove the cOA signal, either once viral infection is cleared, or when the system 'fires' inappropriately due to self-targeting (*Athukoralage et al., 2020b*).

## Crn3 has a highly unusual cooperative catalytic mechanism

*A. fulgidus* Crn3/Csx3 is a much faster enzyme than Crn1. The single turnover reaction rate for $cA_4$ cleavage of 3.6 $min^{-1}$ is closer to that of the AcrIII-1 family, which utilises a distinct active site

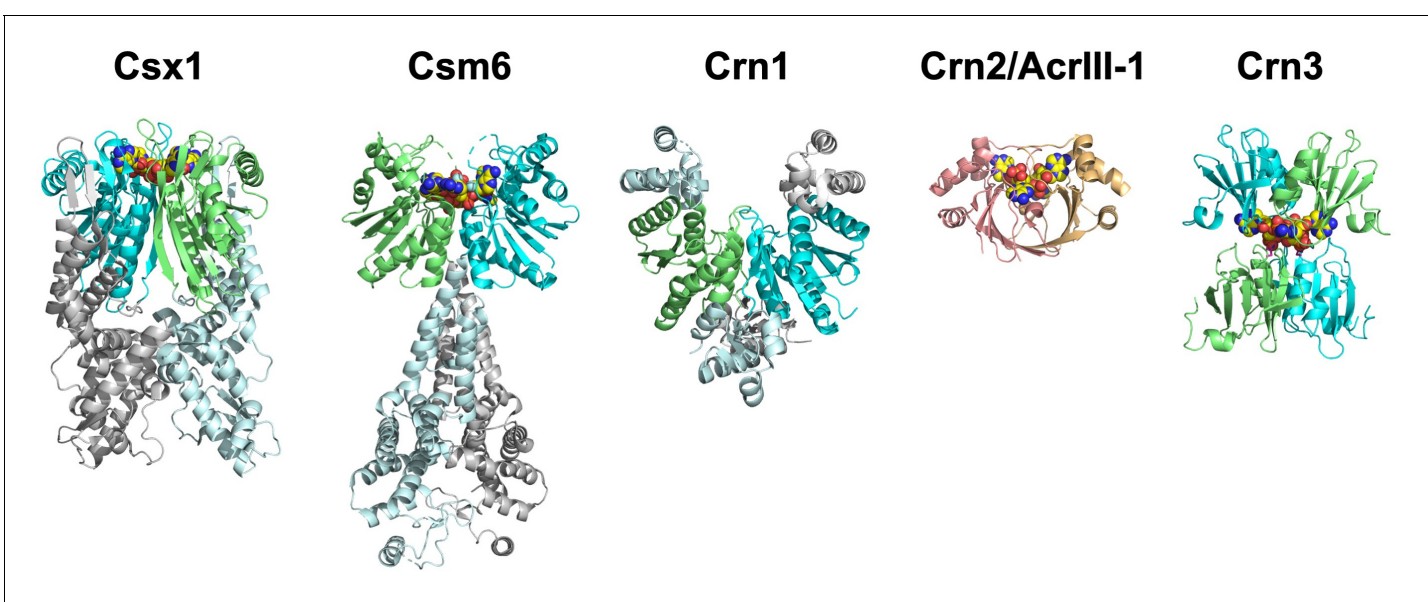

**Figure 7.** The ring nucleases. The Crn1 family (represented here by Sso1393, PDB code 3QYF) is restricted to the crenarchaea (*Athukoralage et al., 2018*). The Csx1 family (represented here by PDB code 6O6Y) is self-limiting $cA_4$-dependent ribonucleases (*Athukoralage et al., 2019*; *Jia et al., 2019*). One Csm6 enzyme (PDB code 6TUG) has been shown to degrade $cA_6$ (*Garcia-Doval et al., 2020*). The Crn3 family is present in type III CRISPR systems in euryarchaeal and cyanobacteria. In contrast, the DUF1874 (Crn2/AcrIII-1) family uses a different protein fold to bind $cA_4$. The Crn2 family of ring nucleases is quite widespread in bacterial type III CRISPR systems (*Samolygo et al., 2020*). The same fold is used by the anti-CRISPR AcrIII-1 in archaeal viruses and bacteriophage (*Athukoralage et al., 2020a*). CARF domains in each ring nuclease (where present) are coloured green and cyan; cOA is shown as yellow spheres.

architecture, employing a conserved histidine residue as a general acid to stabilise the oxyanion leaving group (*Athukoralage et al., 2020a*). This raises an important question: why does Crn3/Csx3 degrade $cA_4$ so quickly when it plays a role in cellular defence rather than viral offense? Clearly, removing a crucial signal of viral infection that mobilises cellular defences is not something that should be undertaken precipitously. A key observation is that Crn3/Csx3 functions via a highly unusual cooperative mechanism where two enzyme dimers associate to sandwich a $cA_4$ substrate molecule. The active site is thus composed of two half-sites that are present on opposite faces of each dimeric moiety, with enzyme tetramers formed transiently to complete the catalytic cycle. There are many examples of allosteric control of enzyme quaternary structure and activity (*Selwood and Jaffe, 2012*) and of active sites shared across subunit interfaces. However, the substrate-induced, non-allosteric, dimerization of Csx3 appears unprecedented in the literature – a partial exception being the example of the Arginine Finger of AAA+ ATPases (*Nagy et al., 2016*), where interdomain interactions with ATP can influence quaternary structure (*Zhao et al., 2016*).

This results in cooperative kinetics where low concentrations of Crn3/Csx3 provide very low levels of ring nuclease activity that rapidly increases with increasing enzyme concentrations. Furthermore, the rate of $cA_4$ turnover is limited by the formation of protein:$cA_4$ complexes or dissociation of the products, rather than the catalytic step, which is significantly faster. Together, these factors may provide a means to control ring nuclease activity in an appropriate manner thus ensuring that $cA_4$ activation of ancillary ribonucleases is allowed to proceed and provide immunity. Thus, alterations in the gene expression levels of Csx3 would have a large effect on the overall rate of catalysis.

Finally, the observed specificity of Crn3/Csx3 for $cA_4$ implicates the corresponding type III CRISPR systems as functioning via a $cA_4$ second messenger. This conforms to the paradigm established from studies of the Crn1 and Crn2/AcrIII-1 family as well as analysis of the specificity of the Csx1/Csm6 ribonucleases, which are most often activated by $cA_4$ (*Grüschow et al., 2019*). Although this conclusion may be influenced by a sampling bias, it appears that $cA_4$ is the default cyclic nucleotide employed to signal infection and activate defences in type III CRISPR systems. It is possible that the pressure applied by $cA_4$-specific anti-CRISPRs has resulted in the utilization of $cA_3$ and $cA_6$ second messengers in some bacterial defence systems. Given the pace of discovery in this area, new cellular and viral enzymes implicated in cyclic nucleotide signalling are anticipated.

## Materials and methods

**Key resources table**

| Reagent type (species) or resource | Designation | Source or reference | Identifiers | Additional information |
|---|---|---|---|---|
| Gene (*Sulfolobus solfataricus*) | SsoCsm complex (eight subunits) | PMID:24119402 | | virus expression construct |
| Gene (*Archaeoglobus fulgidus*) | Csx3/Crn3 | PMID:26106927 | UniProtKB – O28415 | plasmid expression construct |
| Gene (*Methanosarcina mazei*) | MmCsx3 | This paper | UniProtKB - A0A0F8HGG9 | plasmid expression construct |
| Gene (*Mycobacterium tuberculosis*) | MtbCsm complex (five subunits) | PMID:31392987 | | plasmid expression construct |
| Gene (*Thioalkalivibrio sulfidiphilus*) | TsuCsx1 | PMID:31392987 | UniProtKB – B8GSI1 | plasmid expression construct |
| Gene (Thermoanaerobacterium phage THSA-485A) | AcrIII-1 | PMID:31942067 | UniProtKB - I3VYU1 | plasmid expression construct |

### Cloning

For cloning, synthetic genes (g-blocks) encoding *A. fulgidus* or *M. mazei* Csx3 (*Figure 1—figure supplement 1*), codon optimised for expression in *Escherichia coli*, were purchased from Integrated DNA Technologies (IDT), Coralville, USA, and cloned into the pEhisV5spacerTev vector between the NcoI and BamHI restriction sites (*Rouillon et al., 2019*). Competent DH5α (*E. coli*) cells were transformed with the construct and sequence integrity was confirmed by sequencing (Eurofins Genomics). The plasmid was transformed into *E. coli* C43 (DE3) cells for protein expression. The following

constructs were used in plasmid immunity assays: pCsm1-5_ΔCsm6 (*M. tuberculosis csm1-5* under T7 and lac promoter control) and pCRISPR_TetR (CRISPR array with tetracycline resistance gene targeting spacers and *M. tuberculosis cas6* under T7 promoter control) which have been described previously (*Grüschow et al., 2019*); pRAT-Duet constructs containing *T. sulfidiphilus csx1* under arabinose-inducible promoter control with and without viral AcrIII-1 which have been previously described (*Athukoralage et al., 2020b*). *M. mazei csx3* was cloned into the pRAT-Duet_tsuCsx1 vector using the NdeI and XhoI restriction sites. Constructs were verified by sequencing.

## Protein Production and Purification

For expression of *A. fulgidus* or *M. mazei* Csx3, the standard protocol described in detail previously was followed (*Rouillon et al., 2019*). Briefly, a culture was grown in 2L Luria-Broth at 37°C to an $OD_{600}$ of 0.8 AU with shaking at 180 rpm. Protein expression was induced with 0.4 mM isopropyl β-D-1-thiogalactopyranoside and cells were grown at 25°C overnight before harvesting by centrifugation. For protein purification the cell pellet was resuspended in lysis buffer containing 50 mM Tris-HCl 7.5, 0.5 M NaCl, 10 mM imidazole and 10% glycerol supplemented with EDTA-free protease inhibitor tablets (Roche; 1 tablet per 100 ml buffer) and lysozyme (1 mg/ml). Cells were lysed by sonication and the lysate was ultracentrifuged at 40,000 rpm (70 Ti rotor) at 4°C for 35 min. The lysate was loaded onto a 5 ml HisTrap FF Crude column (GE Healthcare) equilibrated with wash buffer containing 50 mM Tris-HCl pH 7.5, 0.5 M NaCl, 30 mM imidazole and 10% glycerol. Unbound protein was eluted with 20 column volumes (CV) of wash buffer prior to elution of His-tagged protein using a step gradient of imidazole (holding at 10% for 3 CV, and 50% for 3 CV) of elution buffer containing 50 mM Tris-HCl pH 7.5, 0.5 M NaCl, 0.5 M imidazole and 10% glycerol. The His-tag was removed by incubating concentrated protein overnight with Tobacco Etch Virus (TEV) protease (1 mg per 10 mg protein) while dialysing in buffer containing 50 mM Tris-HCl pH 7.5, 0.5 M NaCl, 30 mM imidazole and 10% glycerol at room temperature. Cleaved protein was passed through a 5 ml HisTrapFF column and further purified by size-exclusion chromatography (S200 16/60 column; GE Healthcare) in buffer containing 20 mM Tris-HCl pH 7.5, 0.125 M NaCl. After SDS-PAGE, pure protein was pooled, concentrated and stored at −80°C. H60A, D69A, R71 and H80 variants of *A. fulgidus* Csx3 (*Figure 1—figure supplement 1*) were generated using the QuikChange Site-Directed Mutagenesis kit as per manufacturer's instructions (Agilent Technologies) and purified as for the wild-type protein.

## RNA cleavage assays

For RNA cleavage, Csx3 (8 µM protein dimer) was incubated with 50 nM radiolabelled RNA oligonucleotide 49-9A in buffer containing 20 mM Tris-HCl pH 7.5, 150 mM NaCl, 1 mM DTT, three units SUPERase•In Inhibitor and supplemented with 2 mM $MnCl_2$ at 50°C. A control reaction incubating RNA in buffer without protein was also carried out. Reactions were quenched by adding EDTA (17 mM final concentration) and incubated with 20 µg Proteinase K (Invitrogen) for 30 min at 37°C. Subsequently, the reactions were deproteinised by phenol-chloroform extraction and 8 µl reaction product was extracted into 5 µl 100% formamide xylene-cyanol dye. All experiments were carried out in triplicate and RNA cleavage was visualised by phosphor imaging following denaturing polyacrylamide gel electrophoresis (PAGE) as previously described (*Rouillon et al., 2019*).

Oligo 49-9A: 5'-AGGGUACAGUUUGGGUAUUAGCCGUUCUGGUCCUUAUACGAAAAAAAAA.

## Radiolabelled $cA_4$ cleavage assays

Cyclic oligoadenylate (cOA) was generated using *S. solfataricus* Csm complex as detailed in *Rouillon et al., 2019*. Single turnover kinetics experiments for *A. fulgidus* or *M. mazei* Csx3 and variants (8 µM protein dimer) were performed by incubating protein with radiolabelled SsoCsm and cOA (~200 nM $cA_4$) in buffer containing 20 mM Tris-HCl pH 7.5, 150 mM NaCl, 2 mM $MnCl_2$, 1 mM DTT and three units SUPERase•In Inhibitor at 50°C. At desired time points, 10 µl aliquot was removed from the reaction and quenched by adding to phenol-chloroform and vortexing. Products were further isolated by chloroform extraction for thin layer chromatography (TLC). A control reaction incubating cOA in buffer without protein to the endpoint of each experiment was also carried out. All experiments were carried out in triplicate and $cA_4$ degradation was visualised by phosphor imaging. For experiments examining metal dependence, Csx3 was incubated with 1 mM EDTA in

buffer as above before adding cOA and supplementing with 3 mM MgCl$_2$, CaCl$_2$ or CoCl$_2$. Pilot experiments showed that Csx3 was specific for cA$_4$ and did not degrade cA$_6$.

Multiple turnover kinetics of Csx3 was carried out by varying enzyme concentration (240, 320, 640, 800, 960, 1120, 1280, 1920, 2560, 3840, 5120 nM dimer) and incubating with a mix of unlabelled and radiolabelled cA$_4$ (total concentration 128.5 µM) in buffer containing 20 mM Tris-HCl pH 7.5, 150 mM NaCl, 2 mM MnCl$_2$, 1 mM DTT and three units SUPERase•In Inhibitor at 70℃. Reaction products were visualised by phosphor imaging following TLC. All experiments were done in triplicate. Initial rates were calculated and adjusted for labelled:unlabelled cA$_4$ concentration before fitting to the Hill equation on Kaleidagraph (Synergy Software). For activity rescue of Csx3 variants, D69A (4 µM dimer) variant and H60A variant (4 µM dimer) were mixed together and incubated with cA$_4$ (~200 nM) in a reaction supplemented with 2 mM MnCl$_2$ at 70℃. Reactions were quenched at 10, 60 and 600 s by adding phenol-chloroform and vortexing. All experiments were done in triplicate.

TLC was carried out as previously described (*Han et al., 2017b*), by spotting 1 µl of radiolabelled product 1 cm from the bottom of a 20 × 20 cm silica gel TLC plate (Supelco Sigma-Aldrich). TLC was carried out at 35℃ in a humidified chamber with running buffer composed of 30% H$_2$O, 70% ethanol and 0.2 M ammonium bicarbonate, pH 9.2. Sample migration was visualised by phosphor imaging the TLC plate. For kinetic analysis, cA$_4$ cleavage was quantified using the Bio-Formats plugin (*Linkert et al., 2010*) of ImageJ as distributed in the Fiji package (*Schindelin et al., 2012*). The data were fitted to a single exponential curve (y = m1 + m2*(1 - exp(-m3*x))) using Kaleidagraph, as described previously (*Sternberg et al., 2012*).

## Electrophoretic mobility shift assays

Csx3 and variants (0.01, 0.1, 0.5, 1, 10 and 20 µM dimer) were incubated with 20 nM radiolabelled cA$_4$ in binding buffer containing 20 mM Tris-HCl pH 7.5, 150 mM NaCl and 1 mM EDTA supplemented with UltraPure BSA (Thermofisher) for 10 min at 25℃. To examine RNA binding by Csx3, 50 nM 5'-end radiolabelled 3' poly-adenylate tailed RNA oligonucleotide 49-9A was incubated with Csx3 as above. Subsequently, a reaction volume equivalent of 20% glycerol was added to each reaction and native (15% acrylamide, 1X TBE) gel electrophoresis at 30℃ and 250V was performed. Gels were phosphor imaged overnight at −80℃. All experiments were done in triplicate.

## Protein crystallisation

Crystallisation conditions were tested with JCSG and PACT 96 well commercial screens (Jena Biosciences) with Csx3 H60A at a concentration of 13.5 mg/ml. Following optimisation, crystals were obtained from 25% (v/v) Jeffamine M-600 and 100 mM HEPES pH 7.5 using hanging drops in a 24 well plate. 3 µl drops in a 2:1 or 1:1 protein:mother liquor ratio were added to a silanized cover slip over 400 µl mother liquor and sealed with high-vacuum grease (DOW Corning, USA) and left to grow at room temperature. Prior to addition of the cA$_4$ ligand, crystals were harvested into a fresh 2 µl drop of mother liquor and 1 µl of 16 mM cA$_4$ solution was added and left to soak for 12 hr. Crystals were harvested and cryoprotected with the addition of 2 µl 50% (v/v) Jeffamine M-600 in 0.5 µl increments, mounted on loops and vitrified in liquid nitrogen.

## X-ray data processing, structure solution, and refinement

Data on Csx3 H60A crystals soaked with cA$_4$ were collected at Diamond Light Source (DLS) on beamline I04 at a wavelength of 0.9795 Å to 1.84 Å resolution. Diffraction images were automatically processed through the Xia2 pipeline (*Winter, 2010*) using DIALS (*Gildea et al., 2014*) and AIMLESS (*Evans, 2006*). After checking the likely cell content by the Matthews' coefficient, MOLREP (*Vagin and Teplyakov, 1997*) was used to solve the structure using molecular replacement with the Csx3 structure (PDB 3WZI) (*Yan et al., 2015*) as the search model with ligand and water molecules removed. REFMAC5 (*Murshudov et al., 2011*) and Coot (*Emsley and Cowtan, 2004*) were used for automated and manual refinement respectively, which included addition of the ligand and water molecules. cA$_4$ was drawn using Chemdraw (Perkin Elmer) and restraints generated in JLigand (*Lebedev et al., 2012*). The model was corrected and validated using tools in PDB-redo (*Joosten et al., 2014*) and Molprobity (*Chen et al., 2010*). The Molprobity score is 1.04; centile 100. Ramachandran statistics are 97.34% allowed, 0% disallowed. Data processing and refinement

statistics are shown in *Supplementary file 1*. The coordinates and data have been deposited in the Protein Data Bank with accession code 6YUD.

## Dynamic light scattering

Dynamic light scattering experiments were carried out in a 20 µl quartz cuvette using a Zetasizer Nano S90 instrument (Malvern). 80 µM AfCsx3 was either measured alone or when mixed with an equimolar concentration of $cA_4$ in phosphate buffered saline, pH 7.5 at 25°C. Three technical replicates were carried out and by default triplicate measurements were made to produce an average for each technical replicate. For visual inspection of the effect of adding $cA_4$ to AfCsx3 and variants, 80 µM AfCsx3 was added to equimolar $cA_4$ in PBS supplemented with 2 mM $MnCl_2$ in a 100 µl reaction volume and heated to either 25°C or 70°C for 10 min before photographing using a 12-megapixel $f$/1.8-aperture camera.

## Liquid chromatography high-resolution mass spectrometry

Samples were generated by incubating Csx3 (10 µM dimer) with 100 µM synthetic $cA_4$ (BIOLOG Life Science Institute, Bremen) in buffer containing 20 mM Tris-HCl pH 7.5, 150 mM NaCl, 1 mM DTT and 2 mM $MnCl_2$ for 10 min at 50°C. Reactions were quenched by adding EDTA to a final concentration of 25 mM and acidified with trifluoroacetic acid. The acidified sample was bound to a C18 cartridge (Harvard Apparatus), and salts and buffer were washed away with 0.1% trifluoroacetic acid, 2% acetonitrile. Adenylates were eluted from the cartridge with 20 mM ammonium bicarbonate, 50% acetonitrile leaving most protein bound to the resin. Liquid chromatography-high resolution mass spectrometry (LC-HRMS) analysis was performed on a Thermo Scientific Velos Pro instrument equipped with HESI source and Dionex UltiMate 3000 chromatography system as previously described (*Grüschow et al., 2019*). Data were analysed using Xcalibur (Thermo Scientific).

## Plasmid immunity assay comparing viral and host ring nucleases

These assays were performed largely as described previously (*Athukoralage et al., 2020a*). Cells containing *Mycobacterium tuberculosis* (Mtb)Csm1-5, Cas6 and a CRISPR array targeting the tetracycline resistance gene of pRAT-Duet were transformed with the target plasmid containing genes encoding the $cA_4$-activated ancillary nuclease *T. sulfidiphilus* (Tsu) Csx1 and a ring nuclease (*M. mazei* Csx3 or the anti-CRISPR ring nuclease AcrIII-1 from *Thermoanaerobacterium phage THSA-485*). Transformants were allowed to recover on selective LB plates in the presence of 0.2% lactose and 0.02% arabinose (the former for induction of the Cas genes and the ring nuclease, the latter induces the Csx1 plus ring nuclease). The experiment was run with increasing arabinose concentrations (0, 0.002, 0.02, 0.2 % w/v); there was no difference between 0% and 0.002%, and a slight difference between 0.02% and 0.2% arabinose. We did not test directly for toxicity of the Csx3 enzyme in *E. coli*, but judge this to be unlikely due to its specificity for $cA_4$ degradation. Four technical replicates on two biological replicates were carried out, N = 8. Unpaired Welch t test was performed using GraphPad Prism version 8.3.1, GraphPad Software, La Jolla California USA, www.graphpad.com.

## Acknowledgements

We thank Euan Wakefield and Lewis MacDonald for technical assistance with experiments.

## Additional information

### Funding

| Funder | Grant reference number | Author |
| --- | --- | --- |
| Biotechnology and Biological Sciences Research Council | BB/S000313/1 | Sabine Grüschow Shirley Graham Malcolm F White |

| Biotechnology and Biological Sciences Research Council | BB/T004789/1 | Stuart McQuarrie<br>Sabine Grüschow<br>Tracey M Gloster<br>Malcolm F White |
| Wellcome Trust Institutional Strategic Support Fund | 204821/Z/16/Z | Stuart McQuarrie<br>Tracey M Gloster<br>Malcolm F White |

The funders had no role in study design, data collection and interpretation, or the decision to submit the work for publication.

### Author contributions
Januka S Athukoralage, Data curation, Formal analysis, Investigation, Methodology, Writing - original draft, Writing - review and editing; Stuart McQuarrie, Shirley Graham, Investigation, Methodology, Writing - review and editing; Sabine Grüschow, Formal analysis, Investigation, Methodology, Writing - review and editing; Tracey M Gloster, Formal analysis, Supervision, Funding acquisition, Writing - original draft, Project administration, Writing - review and editing; Malcolm F White, Conceptualization, Formal analysis, Funding acquisition, Writing - original draft, Project administration, Writing - review and editing

### Author ORCIDs
Januka S Athukoralage (iD) https://orcid.org/0000-0002-1666-0180
Stuart McQuarrie (iD) http://orcid.org/0000-0003-4828-4842
Shirley Graham (iD) http://orcid.org/0000-0002-2608-3815
Tracey M Gloster (iD) https://orcid.org/0000-0003-4692-2222
Malcolm F White (iD) https://orcid.org/0000-0003-1543-9342

### Decision letter and Author response
Decision letter https://doi.org/10.7554/eLife.57627.sa1
Author response https://doi.org/10.7554/eLife.57627.sa2

## Additional files

### Supplementary files
• Supplementary file 1. Data collection and refinement statistics for H60A mutant of Csx3 in complex with $cA_4$.

• Supplementary file 2. Dynamic light scattering studies with AfCsx3.

• Transparent reporting form

### Data availability
Diffraction data have been deposited in the PDB under the accession code 6YUD.

The following dataset was generated:

| Author(s) | Year | Dataset title | Dataset URL | Database and Identifier |
|---|---|---|---|---|
| Athukoralage JS, McQuarrie S, Grüschow S, Graham S, Gloster TM, White MF | 2020 | Data from: Tetramerisation of the CRISPR ring nuclease Crn3/Csx3 facilitates cyclic oligoadenylate cleavage | https://www.rcsb.org/structure/6YUD | RCSB Protein Data Bank, 6YUD |

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
