## [Decision Letter]

**Acceptance summary:**

Prokaryotic type III CRISPR systems rely on cyclic oligonucleotide second messengers such as cyclic tetraadenylate to coordinate antiviral defense. Here, Athukoralage and colleagues biochemically and structurally characterize Crn3, a new class of cyclic tetraadenylate nuclease. Unexpectedly, the Crn3 active site is formed between a sandwich of two Crn3 dimers, with the opposite face of each dimer contributing essential catalytic residues. Thus, the enzyme displays a highly unusual cooperative cleavage mechanism, which is expected to allow for tight regulatory control.

**Decision letter after peer review:**

Thank you for submitting your article "Tetramerisation of the CRISPR ring nuclease Csx3 facilitates cyclic oligoadenylate cleavage" for consideration by *eLife*. Your article has been favorably reviewed by three peer reviewers, including Joe Wade as the Reviewing Editor and Reviewer #1, and the evaluation has been overseen by John Kuriyan as the Senior Editor. All the reviewers have agreed to reveal their identity: Philip Kranzusch (Reviewer #2); Dipali Sashital (Reviewer #3).

The reviewers have discussed the reviews with one another and the Reviewing Editor has drafted this decision to help you prepare a revised submission.

We would like to draw your attention to changes in our revision policy that we have made in response to COVID-19 (https://elifesciences.org/articles/57162). Specifically, we are asking editors to accept without delay manuscripts, like yours, that they judge can stand as *eLife* papers without additional data, even if they feel that they would make the manuscript stronger. Thus, the revisions requested below only address clarity and presentation.

Prokaryotic type III CRISPR systems rely on cyclic oligonucleotide second messengers such as cA_4_ to coordinate antiviral defense. Here, Athukoralage et al. biochemically and structurally characterize Csx3, a new class of cA_4_ nuclease. Unexpectedly, the Csx3 active site is formed between a sandwich of two Csx3 dimers, with the opposite face of each dimer contributing essential catalytic residues. Thus, the enzyme displays a highly unusual cooperative cleavage mechanism, which is expected to allow for tight regulatory control.

The reviewers were uniformly positive about the paper. The work is of high quality and the conclusions are well supported by the data throughout, and the reviewers were excited about the novel enzyme mechanism involving a pair of Csx3 dimers. No major changes are required. There are just a few comments, listed below, that should be straightforward to address.

Essential revisions:

– The authors mention comparing the apo structure with the cA_4_-bound structure in the third paragraph of the subsection “The structure of Csx3 reveals a head-to-tail filament stabilised by cA_4_”, but there is not a figure depicting the description. An overlay of the two structures would help the reader to visualize the movement of R71 that is described in the text.

– It seems that the Kd of Csx3 for cA_4_ (Figure 1, ~50 nM) is much lower than the half-maximal concentration for cleavage under multiple turnover conditions (Figure 6, ~2000 nM). This may suggest that Csx3 binds cA_4_ with much higher affinity on one face than on the other. If this is the case, is it possible that Csx3 competes with other CARF proteins for binding to cA_4_ even when cellular Csx3 concentrations are too low to allow tetramerization and catalysis?

– In the legend for Figure 1, the authors state an apparent Kd of ~50 nM for cA_4_. Is this based on the observation that there is little binding at 10 nM and nearly complete binding at 100 nM Csx3? Or have the authors performed a binding experiment with a finer range of concentrations between 10-100 nM Csx3? If the former, it may be better to set an upper bound for the Kd at 100 nM (e.g. Kd <100 nM) than to estimate it as the middle of these two values. If the latter, it would be helpful to show an example image of such a binding experiment as a figure supplement.

– It would be helpful if Csx3 structure were shown from the same angle in all figures (Figures 4A, 5A and 7). Similarly, would it be possible to make the structures in Figure 7 the same size with respect to one another, to give the reader a better idea of the relative size/cA_4_ binding sites for each ring nuclease? It would also be helpful to use the same coloring for the cA_4_ and show it in the same orientation for all cA_4_-bound proteins.

– Future reference to the seminal work in this paper will be clearer if the name Crn3 appears in the manuscript title. Consider using Crn3 or both Crn3 / Csx3 in the title of the manuscript.

– The authors describe that cA_4_ was soaked into pre-formed Crn3 crystals. The ability of cA_4_ to diffuse into the pre-formed filamentous crystal packing is surprising and may have important implications for the mechanism of cA_4_ recognition or product release. Did the authors collect X-ray data for apo Crn3 crystals that did not undergo cA_4_ soaking? While these data are not essential for publication, they would be very interesting to help compare movement of residues in the cA_4_ binding pocket related to analysis for Figure 5.

– In the experiments presented in Figure 3, does expression of the ring nuclease alone affect transformation efficiency? This experiment is not required for publication, but it would be useful if the authors can comment if this has been tested.

– Does Csx3 cleave other cyclic oligoadenylate molecules like cA3 or cA6? This experiment is not required for publication, but it would be useful if the authors can comment if this has been tested.

---

## [Author Response]

Essential revisions:– The authors mention comparing the apo structure with the cA_4_-bound structure in the third paragraph of the subsection “The structure of Csx3 reveals a head-to-tail filament stabilised by cA_4_”, but there is not a figure depicting the description. An overlay of the two structures would help the reader to visualize the movement of R71 that is described in the text.

Thank you for pointing out this omission. We have now included this as Figure 5—figure supplement 3.

– It seems that the Kd of Csx3 for cA_4_ (Figure 1, ~50 nM) is much lower than the half-maximal concentration for cleavage under multiple turnover conditions (Figure 6, ~2000 nM). This may suggest that Csx3 binds cA_4_ with much higher affinity on one face than on the other. If this is the case, is it possible that Csx3 competes with other CARF proteins for binding to cA_4_ even when cellular Csx3 concentrations are too low to allow tetramerization and catalysis?

Figure 6 shows a plot of initial velocity versus Csx3 concentration at a fixed concentration of cA_4_, so doesn’t reveal any information on the half maximal concentration of cA_4_ under multiple turnover conditions. Given we don’t know the concentration of Csx3 in the cell, it would be speculative to suggest a potential role as a cA_4_ “sink”. Recent calculations of the concentration of cA_4_ generated in infected cells (Athukoralage et al., 2020) make this unlikely.

– In the legend for Figure 1, the authors state an apparent Kd of ~50 nM for cA_4_. Is this based on the observation that there is little binding at 10 nM and nearly complete binding at 100 nM Csx3? Or have the authors performed a binding experiment with a finer range of concentrations between 10-100 nM Csx3? If the former, it may be better to set an upper bound for the Kd at 100 nM (e.g. Kd <100 nM) than to estimate it as the middle of these two values. If the latter, it would be helpful to show an example image of such a binding experiment as a figure supplement.

We have modified the text as suggested.

– It would be helpful if Csx3 structure were shown from the same angle in all figures (Figures 4A, 5A and 7). Similarly, would it be possible to make the structures in Figure 7 the same size with respect to one another, to give the reader a better idea of the relative size/cA_4_ binding sites for each ring nuclease? It would also be helpful to use the same coloring for the cA_4_ and show it in the same orientation for all cA_4_-bound proteins.

We have swapped the orientation of Figure 4A so it matches Figure 5A. We have revised Figure 7 to include the recent structure of Csm6 bound to cA6, scaled the structures and show them in the same orientation with respect to the cOA molecule, and standardised the colours.

– Future reference to the seminal work in this paper will be clearer if the name Crn3 appears in the manuscript title. Consider using Crn3 or both Crn3 / Csx3 in the title of the manuscript.

We agree and have made this change.

– The authors describe that cA_4_ was soaked into pre-formed Crn3 crystals. The ability of cA_4_ to diffuse into the pre-formed filamentous crystal packing is surprising and may have important implications for the mechanism of cA_4_ recognition or product release. Did the authors collect X-ray data for apo Crn3 crystals that did not undergo cA_4_ soaking? While these data are not essential for publication, they would be very interesting to help compare movement of residues in the cA_4_ binding pocket related to analysis for Figure 5.

We were also surprised that the structure of Csx3 in complex with cA_4_ was obtained following soaking of cA_4_ into apo crystals, and to be honest this is difficult to rationalise. We hypothesise that some flexibility in the interface and movement of sidechains has allowed cA_4_ to bind. We have added some text into the paper to this effect.

– In the experiments presented in Figure 3, does expression of the ring nuclease alone affect transformation efficiency? This experiment is not required for publication, but it would be useful if the authors can comment if this has been tested.

This has been not been tested directly, although we do not expect Csx3 to be toxic in *E. coli* given its specificity. This caveat has been added to the Materials and methods.

– Does Csx3 cleave other cyclic oligoadenylate molecules like cA3 or cA6? This experiment is not required for publication, but it would be useful if the authors can comment if this has been tested.

We have tested Csx3 for cleavage of cA6 and found no activity. This has now been added to the text.